# RECURRENT NEURAL NETWORKS WITH TOP-K GAINS FOR SESSION-BASED RECOMMENDATIONS

## ABSTRACT

RNNs have been shown to be excellent models for sequential data and in particular for session-based user behavior. The use of RNNs provides impressive performance benefits over classical methods in session-based recommendations. In this work we introduce a novel ranking loss function tailored for RNNs in recommendation settings. The better performance of such loss over alternatives, along with further tricks and improvements described in this work, allow to achieve an overall improvement of up to 35% in terms of MRR and Recall@20 over previous session-based RNN solutions and up to 51% over classical collaborative filtering approaches. Unlike data augmentation-based improvements, our method does not increase training times significantly.

## 1  INTRODUCTION

Session-based recommendation is a very common recommendation problem that is encountered in many domains such as e-commerce, classified sites, music and video recommendation. In the session-based setting, past user history logs are typically not available (either because the user is new or not logged-in or not tracked) and recommender systems have to rely only on the actions of the user in the current sessions to provide accurate recommendations. Until recently many of these recommendations tasks were tackled mainly using relatively simple methods such as item-based collaborative filtering (Sarwar et al., 2001) or content-based methods. Recurrent Neural Networks (RNNs) have emerged from the deep learning literature as powerful methods for modeling sequential data. These models have been successfully applied in speech recognition, translation, time series forecasting and signal processing. In recommender systems RNNs have been recently applied to the session-based recommendation setting with impressive results (Hidasi et al., 2016a).

The advantage of RNNs over traditional similarity-based methods for recommendation is that they can effectively model the whole session of user interactions (clicks, views, etc.). By modeling the whole session RNNs can in effect learn the 'theme' of the session and thus provide recommendations with increased accuracy (between 20%-30%) over traditional methods.

RNNs in session-based recommendation have been adapted to the task of recommendation. One of the main objectives in recommendation is to rank items by user preference; i.e. the exact ranking or scoring of items in the tail of the item list (items that the user will not like) is not that important, but it is very important to rank correctly the items that the user will like at the top of the list (first 5, 10 or 20 positions). To achieve this with machine learning one has to typically utilize learning to rank techniques(see e.g. (Burges, 2010)) and in particular ranking objectives and loss functions. The current session-based RNN approaches use ranking loss functions and, in particular, pairwise ranking loss functions. As in most deep learning approaches the choice of a good ranking loss can have a very significant influence on performance. Since deep learning methods need to propagate gradients over several layers and in the case of RNNs 'back in time' over previous steps, to optimize the model parameters, the quality of these gradients originating from the loss function influences the quality of the optimization and the model parameters. Moreover the nature of the recommendation task, which typically entails large output spaces (due to large number of items), poses unique challenges that have to be taken into account as well when designing a proper ranking loss function. We will see that the way this large output space issue is tackled is very crucial in achieving good performance.

In this work we analyze ranking loss functions used in RNNs for session-based recommendations, this analysis leads to a new set of ranking loss functions that increase the performance of the RNN up to 30% over previous commonly used losses without incurring in significant computational overheads. We essentially devise a new class of loss functions that combines learnings from the deep learning and the learning to rank literature. Experimental results on several datasets coming from industry validate these impressive improvements, in terms of Mean Reciprocal Rank (MRR) and Recall@20. With these improvements the difference between RNNs and conventional memory-based collaborative filtering jumps to 51% in terms of MRR and Recall@20 demonstrating the potential that deep learning methods bring to the area of Recommender Systems.

## 1.1 RELATED WORK

One of the main approaches that is employed in session-based recommendation and a natural solution to the problem of a missing user profile is the item-to-item recommendation approach (Sarwar et al., 2001; Linden et al., 2003). In this setting, an item-to-item similarity matrix is precomputed from the available session data, that is items that are often clicked together in sessions are deemed to be similar. This similarity matrix is then simply used during the session to recommend the most similar items to the one the user has currently clicked.

Long Short-Term Memory (LSTM) Hochreiter & Schmidhuber (1997) networks are a type of RNNs that have been shown to solve the optimization issues the plague vanilla-type RNNs. LSTM's include additional gates that regulate when and how much of the input to take into account and when to reset the hidden state. A slightly simplified version of LSTM – that still maintains all their properties – are Gated Recurrent Units (GRUs) Cho et al. (2014), which we use in this work. Recurrent Neural Networks have been used with success in the area of session-based recommendations; (Hidasi et al., 2016a) proposed a Recurrent Neural Network with a pairwise ranking loss for this task, (Tan et al., 2016) proposed data augmentation techniques to improve the performance of the RNN for session-based recommendations; these techniques have though the side effect of increasing training times as a single session is split into several sub-sessions for training. Session-based RNNs have been augmented (Hidasi et al., 2016b) with feature information, such as text and images from the clicked/consumed items, showing improved performance over the plain models. RNNs have also been used in more standard user-item collaborative filtering settings where the aim is to model the evolution of the user and items factors (Wu et al., 2017),(Devooght & Bersini, 2016) where the results are less striking, with the proposed methods barely outperforming standard matrix factorization methods. This is to be expected as there is no strong evidence on major user taste evolution in a single domain in the timeframes of the available datasets and sequential modeling of items that are not 'consumed' in sessions such as movies might not bring major benefits.

Another area touched upon in this work are loss functions tailored to recommender systems requirements. This typically means ranking loss functions. In this area there has been work particularly in the context of matrix factorization techniques. One of the first learning to rank techniques for collaborative filtering was introduced in (Weimer et al., 2007). Essentially a listwise loss function was introduced along with an alternating bundle method for optimization of the factors. Further ranking loss function for collaborative filtering were introduced in (Shi et al., 2012) (Rendle et al., 2009b) and (Koren & Sill, 2011). Note that the fact that these loss functions work well in matrix factorization does not guarantee in any way that they are an optimal choice for RNNs as backpropagation requirements are stronger than those posed by simple SGD. We will in fact see that BPR, a popular choice of loss function, needs to be significantly modified to extract optimal results in the case of RNNs for session-based recommendations. Another work related to sampling large output spaces in deep networks for efficient loss computations for language models is the 'blackout' method (Ji et al., 2016), where essentially a sampling procedure similar to the one used in (Hidasi et al., 2016a) is applied in order to efficiently compute the categorical cross-entropy loss.

## 2 SAMPLING THE OUTPUT

In the remainder of the paper we will refer to the RNN algorithm implemented in (Hidasi et al., 2016a) as GRU4Rec, the name of the implementation published by the authors on github [1]. In this

---

[1] https://github.com/hidasib/GRU4Rec

section we revisit how GRU4Rec samples negative feedback on the output and discuss its importance. We extend this sampling with an option for additional samples and argue that this is crucial for the increased recommendation accuracy we achieve (up to 51% improvement).

In each training step, GRU4Rec takes the item of the current event in the session – represented by a one-hot vector – as an input. The output of the network is a set of scores over the items, corresponding to their likelihood of being the next item in the session. The training iterates through all events in the sequence. The complexity of the training with backpropagation through time is $O(N_E(H^2 + HN_O))$ where $N_E$ is the number of training events, $H$ is the number of hidden units and $N_O$ is the number of outputs, for which scores are computed. Computing scores for all items is very impractical, since it makes the network unscalable[2]. Therefore GRU4Rec uses a sampling mechanism and during training computes the scores for a subset of the items only.

Instead of making a forward and backward pass with one training example only and then moving to the next, the network is fed with a bundle of examples and is trained on the mean gradient. This common practice is called *mini-batch training* and has several benefits, e.g. utilizing the parallelization capabilities of current hardware better, thus training faster, and producing more stable gradients than stochastic gradient training and thus converging faster. GRU4Rec introduced mini-batch based sampling Hidasi et al. (2016a). For each example in the mini-batch, the other examples of the same mini-batch serve as negative examples (see Figure 1).[3] This method is practical from an implementation point of view and can be also implemented efficiently for GPUs.

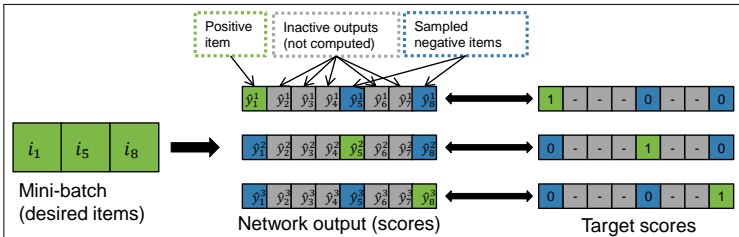

Figure 1: Mini-batch sampling.

The network can be trained with one of three different listwise ranking loss functions (see Section 3). All loss functions require a score for the target item (i.e. for the item which was the actual next item) and score(s) for at least one negative sample (i.e. item other than the target). One property of ranking losses is that learning happens only if the score of the target item does not exceed that of the negative samples by a large margin, otherwise the items are already in the right order, so there is nothing to be learned. Therefore, when utilizing a sampling procedure, it is crucial that high scoring items make it among the negative samples. Whether an item has a high score, depends on the context (item sequence) the scores are actually computed for. Popular items generally score high in many situations, making popularity-based sampling a good sampling strategy. Mini-batch sampling is basically a form of popularity-based sampling, since the training iterates through all events, thus the probability of an item acting as a negative sample is proportional to its support. The problem with popularity-based sampling is that learning can slow down after the algorithm learns to (generally) rank target items above popular ones, and thus can still be inaccurate with ranking long tail high scoring items. On the other hand, uniform sampling slows down learning, due to the high number of low scoring negative samples, but might produce an overall more accurate model if trained indefinitely. In our experience, popularity-based sampling generally produces better results.

Tying sampling to the mini-batches has several practical benefits, but is too restrictive for three reasons. (1) Mini-batch sizes are generally small, ranging from few tens to few hundreds. If the number of items is large, the small sample size further hinders the chance of including all of the high scoring negative examples. (2) Mini-batch size has a direct effect on the training. E.g. we found

---

[2]While it can still result in an acceptable training time for smaller datasets, especially if the number of items is only a few tens of thousand, algorithms scaling with the product of the number of events and items cannot scale up for larger datasets

[3]e.g.: Assume a mini-batch of 32 examples, with one desired output (target) for each example. Scores are computed for all 32 items for each of the 32 examples resulting in $32 \times 32 = 1024$ scores. Thus we have 31 scores of negative examples for each of the targets.

that training with smaller mini-batch sizes (30-100) produces more accurate models, but training with larger ones is faster on the GPU due to parallelization. (3) The sampling method is inherently popularity-based, which generally is a good strategy, but might not be optimal for all datasets.

Therefore we extend the sampling of GRU4Rec with additional samples. We sample $N_A$ items which are shared by the examples of the mini-batch, i.e. the same samples are used for each example[4]. These additional samples are used along with the $N_B - 1$ samples coming from the mini-batch (popularity) sampling. Additional samples can be sampled in any way, we chose to sample proportional to $\text{supp}_i^\alpha$, where $\text{supp}_i$ is the support of the item and $\alpha$ is the parameter of the sampling. $\alpha = 0$ and $\alpha = 1$ gives uniform and popularity-based sampling respectively.

Adding more samples naturally increases the complexity, since $N_O$ increases from $N_B$ to $N_A + N_B$. However, the computations are easily parallelizable, thus there is no actual increase in the training time on modern GPUs up to a certain sample size (see Section 4.1). The efficient implementation of this sampling however is not trivial. Sampling according to a distribution on GPUs is slow, thus it should be handled by the CPU. The sampled item IDs can be given to the GPU along with the item IDs of the mini-batch. Sampling the distribution takes some time every time a new mini-batch is formed, thus GPU execution is frequently interrupted, making GPU utilization low and thus training slow. On the top of that, sampling a few items at once is less efficient than sampling lots of them, even on CPU. Therefore we implemented a cache that pre-samples and stores lots of negative samples. Training uses up these samples and the cache is recomputed once it is empty. We found that pre-sampling 10-100 million item IDs significantly improves training speed when compared to using no cache at all.

## 3 LOSS FUNCTION DESIGN

In this section we examine the loss functions implemented in GRU4Rec and identify their weaknesses. We propose two ways to stabilize the numerical instability of the cross-entropy loss, we show how learning with the TOP1 and BPR pairwise losses degrades as we add more samples to the output, and propose a family of loss functions based on pairwise losses that alleviates this problem. We note that, while our aim is to improve GRU4Rec, the loss functions proposed in this section can be also used with other models, such as matrix factorization.

### 3.1 CATEGORICAL CROSS-ENTROPY

Categorical cross-entropy measures the distance of a proposed (discrete) probability distribution $q$ from the target distribution $p$ as defined by (1).

$$H(p, q) = -\sum_{j=1}^{N} p_j \log q_j \qquad (1)$$

This loss is often used in machine learning and deep learning in particular for multi-class classification problems. Next item recommendation can be interpreted as classification, where the class labels are the items in the system and item sequences need to be assigned with the label of the item that follows. In a single-label scenario – such as next item recommendation – the target distribution is a one-hot vector over the set of items, with the coordinate corresponding to the target item set to 1. The proposed distribution consists of the scores assigned to the items by the algorithm. The output scores need to be transformed to form a distribution. It is common practice to use the *softmax* transformation (2), which is a continuous approximation of the *max* operation. This naturally aligns with the sentiment that the label with the highest score is assigned to the sequence.

$$s_i = \frac{e^{r_i}}{\sum_{j=1}^{N} e^{r_j}} \qquad (2)$$

---

[4]However, the scores of these samples will be still different per example, because of the differing item sequences they are based on.

Cross-entropy in itself is a pointwise loss, (that is it can be computed per individual item) as it is the sum of independent losses defined over the coordinates. Combining it with softmax introduces listwise properties into the loss, since the loss now cannot be separated over coordinates (or items). Putting them together we get the following loss function over the scores (assuming that the target item is indexed by $i$):

$$L_{\text{xe}} = -\log s_i = -\log \frac{e^{r_i}}{\sum_{j=1}^{N} e^{r_j}} \tag{3}$$

**Fixing the instability:** One of the losses available in GRU4Rec was cross-entropy with softmax scores. Hidasi et al. (2016a) reported slightly better results than with other losses, but deemed the loss to be unstable for a large fraction of the hyperparameter space and thus advised against its use. This instability comes from the limited numerical precision. Assuming that there is a $k$ for which $r_k \gg r_i$, $s_i$ becomes very small and rounded to 0, because of the limited precision. The loss then computes $\log 0$, which is undefined. Two ways to circumvent this problem are as follow: (a) compute $-\log(s_i + \epsilon)$, where $\epsilon$ is a very small value (we use $10^{-24}$); (b) compute $-\log s_i$ directly as $-r_i + \log \sum_{j=1}^{N} e^{r_j}$. The former introduces some noise, while the latter does not allow the separated use of the transformation and the loss, but both methods stabilize the loss. We did not observe any differences in the results of the two variants.

## 3.2 RANKING LOSSES: TOP1 & BPR

GRU4Rec offers two loss functions based on pairwise losses. Pairwise losses compare the score of the target to a negative example (i.e. any item other than the target). The loss is high if the target's score is higher than that of the negative example. GRU4Rec computes scores for multiple negative samples per each target, and thus the loss function is composed as the average of the individual pairwise losses. This results in a listwise loss function, which is composed of pairwise losses.

One of the loss functions is coined TOP1 (4). It is a heuristically put together loss consisting of two parts. The first part aims to push the target score above the score of the samples, while the second part lowers the score of negative samples towards zero. The latter acts as a regularizer, but instead of constraining the model weights directly, it penalizes high scores on the negative examples. Since all items act as a negative score in one training example or another, it generally pushes the scores down.

$$L_{\text{top1}} = \frac{1}{N_S} \sum_{j=1}^{N_S} \sigma(r_j - r_i) + \sigma(r_j^2) \tag{4}$$

$j$ runs over the ($N_S$) sampled negative ('non-relevant') items, relevant items are index by $i$. The other loss function (5) is based on the popular Bayesian Personalized Ranking (BPR) Rendle et al. (2009a) loss. Here the negative log-probability of the target score exceeding the sample scores is minimized (i.e. the probability of target scores being above sample scores is maximized). The non-continuous $P(r_i > r_j)$ is approximated by $\sigma(r_i - r_j)$.

$$L_{\text{bpr}} = -\frac{1}{N_S} \sum_{j=1}^{N_S} \log \sigma(r_i - r_j) \tag{5}$$

### 3.2.1 VANISHING GRADIENTS

Taking the average of individual pairwise losses has an undesired side effect. Examining the gradients for the TOP1 and BPR losses w.r.t. the target score $r_i$, ((6) and (7) respectively) reveals that under certain circumstances gradients vanish and thus learning stops. With pairwise losses, one generally wants to have negative samples with high scores, as those samples produce high gradients. Or intuitively, if the score of the negative sample is already well below that of the target, there is nothing to learn from that negative sample anymore. For this discussion we will denote samples where $r_j \ll r_i$ *irrelevant*. For an irrelevant sample $\sigma(r_j - r_i)$ in ((6) and $1 - \sigma(r_i - r_j)$ (7)

will be close to zero. Therefore, any irrelevant sample adds basically nothing to the total gradient. Meanwhile the gradient is always discounted by the total number of negative samples. By increasing the number of samples, the number of irrelevant samples increases faster than that of including relevant samples, since the majority of items is irrelevant as a negative sample. This is especially true for non-popularity-based sampling and high sample numbers. Therefore these losses start to vanish as the number of samples increase, which is counterintuitive and hurts the full potential of the algorithm.[5][6]

$$\frac{\partial L_{\text{top1}}}{\partial r_i} = -\frac{1}{N_S} \sum_{j=1}^{N_S} \sigma(r_j - r_i) \left(1 - \sigma(r_j - r_i)\right) \tag{6}$$

$$\frac{\partial L_{\text{bpr}}}{\partial r_i} = -\frac{1}{N_S} \sum_{j=1}^{N_S} \left(1 - \sigma(r_i - r_j)\right) \tag{7}$$

Note, that TOP1 is sensitive to relevant examples where $r_j \gg r_i$, which is an oversight in the design of the loss. While this is unlikely to happen, it cannot be outruled. For example, when comparing a niche target to a very popular sample – especially during the early phase of learning – the target score might be much lower than the sample score.

We concentrated on the gradients w.r.t. the target score, but a similar issue can be observed for the gradients on the negative scores. The gradient w.r.t. the score of a negative sample is the gradient of the pairwise loss between the target and the sample divided by the number of negative samples. This means that even if all negative samples would be relevant, their updates would still diminish as their number grows.

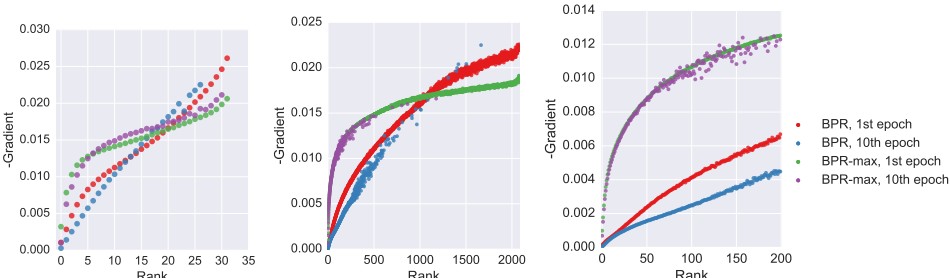

Figure 2: Median negative gradients of BPR and BPR-max w.r.t. the target score against the rank of the target item. Left: only minibatch samples are used (minibatch size: 32); Center: 2048 additional negative samples were added to the minibatch samples; Right: same setting as the center, focusing on ranks 0-200.

## 3.3 RANKING-MAX LOSS FUNCTION FAMILY

To overcome the vanishing of gradients as the number of samples increase, we propose a new family of listwise loss functions, based on individual pairwise losses. The idea is to have the target score compared with the most relevant sample score, which is the maximal score amongst the samples. The general structure of the loss is described by (8).

$$L_{\text{pairwise}-\max}\left(r_i, \{r_j\}_{j=1}^{N_S}\right) = L_{\text{pairwise}}(r_i, \max_j r_j) \tag{8}$$

---

[5]Simply removing the discounting factor does not solve this problem, since it is equivalent of multiplying the learning rate by $N_S$. This would destabilize learning due to introducing high variance into the updates.

[6]For BPR, there is the option of maximizing the sum of individual pairwise probabilities $\sum_{j=1}^{N_S} P(r_i > r_j)$, i.e. minimizing $-\log \sum_{j=1}^{N_S} \sigma(r_i - r_j)$. However, this loss has even worse properties.

The maximum selection is non-differentiable and thus cannot be used with gradient descent. Therefore we use the softmax scores to preserve differentiability. Here, the softmax transformation is only used on the negative examples (i.e. $r_i$ is excluded), since we are looking from the maximum score amongst the negative examples. This naturally results in loss functions where each negative sample is taken into account proportional to its likelihood of having the maximal score. Based on this general idea, we now derive the TOP1-max and BPR-max loss functions.

**TOP1-max:** The TOP1-max loss is fairly straightforward. The regularizing part does not necessarily need to be only applied for the maximal negative score, however we found that this gave the best results, thus kept it this way. The continuous approximation to the maximum selection entails summing over the individual losses weighted by the corresponding softmax scores $s_j$, giving us the TOP1-max loss (9).

$$L_{\text{top1−max}} = \sum_{j=1}^{N_S} s_j \left( \sigma(r_j - r_i) + \sigma(r_j^2) \right) \tag{9}$$

The gradient of TOP1-max (10) is the softmax weighted average[7] of individual pairwise gradients. If $r_j$ is much lower than the maximum of negative scores, its weight will be almost zero and more weight will be placed on examples with scores close to the maximum. This solves the issue of vanishing gradients with more samples, because irrelevant samples will be just ignored, while the gradient will point towards the gradient of the relevant samples. Of course, if all samples are irrelevant, the gradient becomes near zero, but this is not a problem, since if the target score is greater than all sample scores, there is nothing to be learned. Unfortunately, the sensitivity to large sample scores of TOP1 is still an issue as it is the consequence of the pairwise loss and not the aggregation.

$$\frac{\partial L_{\text{top1−max}}}{\partial r_i} = -\sum_{j=1}^{N_S} s_j \sigma(r_j - r_i) \left( 1 - \sigma(r_j - r_i) \right) \tag{10}$$

**BPR-max:** Going back to the probability interpretation of BPR, the goal is to maximize the probability of the target score being higher than the maximal sample score $r_{\max} = \max_j r_j$. This can be rewritten using conditional probabilities:

$$P(r_i > r_{\max}) = \sum_{j=1}^{N_S} P(r_i > r_j | r_j = r_{\max}) P(r_j = r_{\max}) \tag{11}$$

$P(r_i > r_j)$ and $P(r_j = r_{\max})$ is approximated by $\sigma(r_i - r_j)$ (as in the original BPR loss) and the softmax score $s_j$ respectively. We then want to minimize the negative log-probability, which gives us the loss:

$$L_{\text{bpr−max}} = -\log \sum_{j=1}^{N_S} s_j \sigma(r_i - r_j) \tag{12}$$

The gradient of BPR-max (13) is the weighted average of individual BPR gradients, where the weights are $s_j \sigma(r_i - r_j)$. The relative importance of negative samples $j$ and $k$ is $\frac{\sigma(r_i - r_j) s_j}{\sigma(r_i - r_k) s_k} = \frac{e^{r_j} + e^{-r_i + r_j + r_k}}{e^{r_k} + e^{-r_i + r_j + r_k}}$, which behaves like softmax weights if $r_i \gg r_j + r_k$ or if both $r_i$ and $r_k$ are small. Otherwise it is a smoothed softmax. This means that while $r_i$ is small, the weights are distributed more evenly, yet clear emphasis will be given to higher sample scores. As $r_i$ becomes higher, the focus shifts quickly to the samples with high scores. This is an ideal behaviour.

$$\frac{\partial L_{\text{bpr−max}}}{\partial r_i} = -\frac{\sum_{j=1}^{N_S} s_j \sigma(r_i - r_j) \left( 1 - \sigma(r_i - r_j) \right)}{\sum_{j=1}^{N_S} s_j \sigma(r_i - r_j)} \tag{13}$$

---

[7] $\sum s_j = 1$

The gradient w.r.t. a negative sample – with both the BPR-max and TOP1-max – is proportional to the softmax score of the example, meaning that only the items, near the maximum will be updated. This is beneficial, because if the score of a negative sample is low, it doesn't need to be updated. If the score of a sample is much higher than that of the others it will be the only one updated and the gradient will coincide with the gradient of the pairwise loss between the target and the sample score. In a more balanced setting the gradient is between the aforementioned gradient and 0. For example the gradient of BPR-max w.r.t. a negative sample's score is as follows:

$$\frac{\partial L_{\text{bpr-max}}}{\partial r_k} = s_k - \frac{s_k \sigma^2(r_i - r_k)}{\sum_{j=1}^{N_S} s_j \sigma(r_i - r_j)} \tag{14}$$

Figure 2 depicts how the gradients of BPR and BPR-max behave given the rank of the target item[8]. The rank of the target is the number of negative scores exceeding it, e.g. rank 0 means that the target score is higher than all sample scores. Lower rank means that there are fewer negative samples that are relevant. The figure depicts the median negative gradient w.r.t. the target score in two cases, measured on a dataset sample during the $1^{st}$ and $10^{th}$ epochs (i.e. beginning and end of the training): (left) no additional samples were used, only the other examples from a mini-batch of size 32; (middle & right) 2048 additional negative samples were added. The rightmost figure focuses on the first 200 ranks of the figure in the middle. The gradient is slightly higher for BPR when there are more relevant samples (i.e. high ranks). This is natural, since BPR-max focuses on samples closest to the maximum value and ignores other still relevant samples. This entails slightly slower learning for BPR-max when the target item is ranked at the end of the list, but the difference is not really significant. On the other hand, the gradient of BPR quickly vanishes as the number of relevant samples decrease (i.e. low ranks). The point of vanishing is relative to the total sample size. With small sample size, BPR's gradient starts vanishing around rank 5 (the BPR-max does not vanish until rank 0); meanwhile, with more samples, the BPR gradient is very low, even for rank 100-500 (again, the gradient BPR-max starts decreasing significantly later). This means that BPR can hardly push target scores up in the ranking after a certain point, which comes earlier as the number of sample size increases. BPR-max, on the other hand, behaves well and is able to improve the score all the way.

### 3.3.1 BPR-MAX WITH SCORE REGULARIZATION

Even though we showed that the heuristic TOP1 loss is sensitive to relevant samples with very high scores, it was found to be performing better than BPR in Hidasi et al. (2016a). According to our observation, the same is true for the relation of TOP1-max and BPR-max. Part of the reasons lies in the rare occurrence of $r_j \gg r_i$ while $r_j \approx 0$ simultaneously. If only the first condition is met, the gradient w.r.t. $r_i$ might vanish, but the regularizing part of TOP1 makes sure that $r_j$ is moved towards zero, which might even make the update possible for $r_i$ next time (e.g. if $r_j$ was negative, moving it towards zero decreases the difference with $r_i$). The score regularization in TOP1 is very beneficial to the overall learning process, so even though the loss might not be theoretically optimal, it can achieve good results. GRU4Rec support two forms of regularization with every loss: dropout and $\ell_2$ regularization of the model parameters. The regularization of TOP1 is used on the top of these. According to our experiments, the $\ell_2$ regularization of model parameters decreases the model performance. Our assumption is that some of the model weights – such as the weight matrices for computing the update and reset gate – should not be regularized. Penalizing high output scores takes care of constraining the model, even without explicitly regularizing the weights.

Therefore we added score regularization to the BPR-max loss function as well. We tried several ways of score regularization. In the best performing one we conditioned the sample scores on independent, zero mean Gaussians with variance inversely proportional to the softmax score (15). This entails stronger regularization on scores closer to the maximum, which is ideal in our case.

$$P\left(r_i > r_{\max} | \{r_j\}_{j=1}^{N_S}\right) \prod_{j=1}^{N_S} P(r_j) = P\left(r_i > r_{\max} | \{r_j\}_{j=1}^{N_S}\right) \prod_{j=1}^{N_S} \mathcal{N}\left(0, \frac{c}{s_j}\right) \tag{15}$$

---

[8]Similar trends can be observed when comparing TOP1 and TOP1-max, even though the shape of the curves is quite different from that of BPR.

We minimize the negative log-probability and do continuous approximations as before, resulting in the final form of the BPR-max loss function (16). The regularization term is a simple, softmax weighted $\ell_2$ regularization over the scores. $\lambda$ is the regularization hyperparameter of the loss.

$$L_{\mathrm{bpr-max}} = -\log \sum_{j=1}^{N_S} s_j \sigma(r_i - r_j) + \lambda \sum_{j=1}^{N_S} s_j r_j^2 \tag{16}$$

## 4 EXPERIMENTS

**Experimental setup:** We evaluated the proposed improvements – fixed cross-entropy loss, ranking-max loss functions & adding additional samples – on four dataset. RSC15 is based on the dataset of RecSys Challange 2015[9], which contains click and buy events from an online webshop. We only kept the click data. VIDEO and VIDXL are proprietary datasets containing watch events from an online video service. Finally, CLASS is a proprietary dataset containing item page view events from an online classified site. Datasets were subjugated to minor preprocessing then split into train and test sets so that a whole session either belongs to the train or to the test set. The split is based on the time of the first event of the sessions. The datsets and the split are exactly the same for RSC15 as in Hidasi et al. (2016a); and for VIDXL and CLASS as in Hidasi et al. (2016b). VIDEO is of the same source as in Hidasi et al. (2016a), but a slightly different subset. Table 1 overviews the main properties of the datasets.

Table 1: Properties of the datasets.

| Data | Train set | | Test set | | Items |
|------|-----------|--------|----------|--------|-------|
|      | Sessions | Events | Sessions | Events | |
| RSC15 | 7,966,257 | 31,637,239 | 15,324 | 71,222 | 37,483 |
| VIDEO | 2,144,930 | 10,214,429 | 29,804 | 153,157 | 262,050 |
| VIDXL | 17,419,964 | 69,312,698 | 216,725 | 921,202 | 712,824 |
| CLASS | 1,173,094 | 9,011,321 | 35,741 | 254,857 | 339,055 |

Evaluation is done under the next item prediction scenario, that is we iterate over test sessions and events therein. For each event, the algorithm guesses the item of the next event of that session. Since the size of the VIDXL test set is large, we compare the target item's score to that of the 50,000 most popular items during testing, similarly to Hidasi et al. (2016b). While this evaluation for VIDXL overestimates the performance, the comparison of algorithms remain fair Bellogin et al. (2011). As recommender systems can only recommend a few items at once, the actual item a user might pick should be amongst the first few items of the list. Therefore, our primary evaluation metric is recall@20 that is the proportion of cases having the desired item amongst the top-20 items in all test cases. Recall does not consider the actual rank of the item as long as it is amongst the top-N. This models certain practical scenarios well where there is no highlighting of recommendations and the absolute order does not matter. Recall also usually correlates well with important online KPIs, such as click-through rate (CTR)Liu et al. (2012); Hidasi & Tikk (2012). The second metric used in the experiments is MRR@20 (Mean Reciprocal Rank). That is the average of reciprocal ranks of the desired items. The reciprocal rank is set to zero if the rank is above 20. MRR takes into account the rank of the item, which is important in cases where the order of recommendations matter (e.g. the lower ranked items are only visible after scrolling).

The natural baseline we use is the original GRU4Rec algorithm, upon which we aim to improve. We consider the results with the originally proposed TOP1 loss and *tanh* activation function on the output to be the baseline. The hidden layer has 100 units. We also indicate the performance of item-kNN, a natural baseline for next item prediction. Results for RSC15, VIDXL and CLASS are taken directly from corresponding papers Hidasi et al. (2016a;b) and measured with the optimal hyperparameters in Hidasi et al. (2016a) for VIDEO. We do separate hyperparameter optimization on a separate validation set for the proposed improvements.

---

[9]http://2015.recsyschallenge.com

The methods are implemented under the Theano framework Al-Rfou et al. (2016) in python. Experiments were run on various GPUs, training times were measured on an unloaded Titan X (Maxwell) GPU. Code is available publicly on GitHub[10] for reproducibility.

## 4.1 USING ADDITIONAL SAMPLES

The first set of experiments examines the effect of additional negative samples on recommendation accuracy. Experiments were performed on the CLASS and the VIDEO datasets. Since results are quite similar we excluded the VIDEO results to save some space. Figure 3a depicts the performance of the network with TOP1, cross-entropy, TOP1-max and BPR-max losses. Recommendation accuracy was measured with different number of additional samples, as well as in the case when all scores are computed and there is no sampling. As we discussed earlier, this latter scenario is a more theoretical one, because it is not scalable. As theory suggests (see Section 3), the TOP1 loss does not cope well with lots of samples. There is a slight increase in performance with a few extra samples, as the chance of having relevant samples increases; but performance quickly degrades as sample size grow, thus lots of irrelevant samples are included. On the other hand, all three of the other losses react well to adding more samples. The point of diminishing return is around a few thousand of extra samples for cross-entropy. TOP1-max starts to slightly lose accuracy after that. BPR-max improves with more samples all the way, but slightly loses accuracy when all items are used.

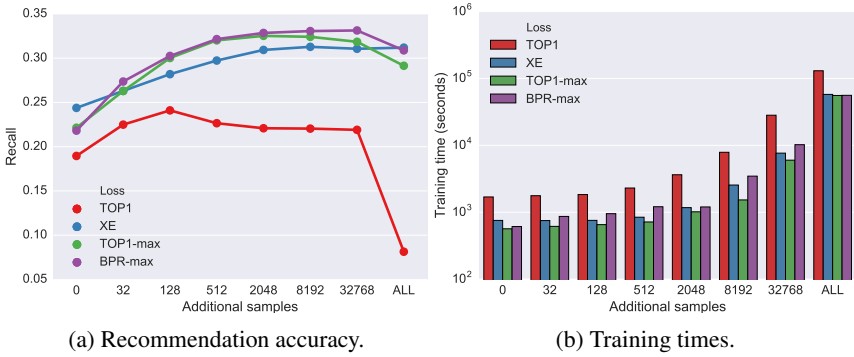

(a) Recommendation accuracy.          (b) Training times.

Figure 3: Results on the CLASS dataset. "ALL" means no sampling of items.

Adding extra samples increase computational cost, yet due to easy parallelization on modern GPUs most of this cost is alleviated. Figure 3b shows the training times at different sample sizes. Please note the logarithmic scale. The actual training time depends on not just the dataset, but model parameters (especially mini-batch size) and how certain operators used for computing the loss are supported by the framework. The trend, however, is similar to for all losses. For example, the full training of the network is around 10 minutes (with the settings for cross-entropy or TOP1-max), which does not increase with even 512 extra samples. At the point of diminishing returns, i.e. at 2048 extra samples, training time is around 15 minutes, which is also totally acceptable. After that, training times grow quickly, due to exceeding the parallelization capabilities of the GPU we used. The trend is similar on the VIDEO dataset, with training times starting around 50 minutes, starting to increase at 2048 extra samples (to 80 minutes) and quickly above thereafter. This means that the proposed method can be used with zero too little additional cost in practice, unlike data augmentation methods. It is also clear that GRU4Rec can work just as well with a few thousands of negative examples as with the whole itemset, thus it can be kept scalable.

In the next experiment we perform a parameter sensitivity analysis of the $\alpha$ parameter that controls the sampling. Figure 4 depicts the performance over different $\alpha$ values for the cross-entropy, TOP1-max and BPR-max losses. Cross-entropy favors higher $\alpha$ values with low sample sizes and low $\alpha$ values for large samples. This is inline with our discussion in Section 2: popular samples are useful when the sample size is very limited and at the beginning of the training, but might be exhausted quickly, thus switching to a more balanced sampling can be beneficial if we have the means to (e.g. large enough sample size). Also, the uniform sampling in this case is supplemented by the few

---

[10]https://github.com/blinded

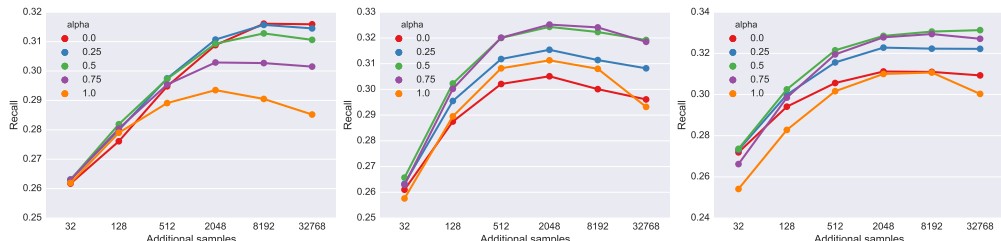

Figure 4: The effect of the alpha parameter on recommendation accuracy at different sample sizes on the CLASS dataset. Left: cross-entropy loss; Middle: TOP1-max loss; Right: BPR-max loss.

popularity based samples of the mini-batch sampling. The ranking-max losses, on the other hand, seem to prefer the middle road with a slight preference towards higher values, while the extremes perform the worst. We assume that this is mostly due to (a) being based on pairwise losses, where popular samples are usually desired; (b) and the score regularization: with popularity based sampling the scores of the most popular items would be decreased beyond what is desirable.

Table 2: Recommendation accuracy with additional samples and different loss functions compared to item-kNN and the original GRU4Rec. Improvements over item-kNN and the original GRU4Rec (with TOP1 loss) results are shown in parentheses. Best results are typeset bold.

| Dataset | Item kNN | GRU4Rec original | XE | TOP1 | GRU4Rec with additional samples XE | TOP1-max | BPR-max |
|---|---|---|---|---|---|---|---|
| | | | | | *Recall@20* | | |
| RSC15 | 0.5065 | 0.5853 | 0.5781 | 0.6117 (+20.77%, +4.51%) | 0.7112 (+40.41%, +21.51%) | 0.7086 (+39.91%, +21.07%) | **0.7190 (+41.95%, +22.84%)** |
| VIDEO | 0.5201 | 0.5051 | 0.5060 | 0.5325 (+2.40%, +5.43%) | 0.6222 (+19.63%, +23.18%) | 0.6421 (+23.46%, +27.12%) | **0.6524 (+25.44%, +29.16%)** |
| VIDXL | 0.6263 | 0.6831 | 0.7046 | 0.6723 (+7.35%, -1.58%) | 0.7972 (+27.29%, +16.70%) | 0.7935 (+26.70%, +16.16%) | **0.8020 (+28.05%, +17.41%)** |
| CLASS | 0.2201 | 0.2478 | 0.2545 | 0.2342 (+6.41%, -5.50%) | 0.3099 (+40.83%, +25.07%) | 0.3252 (+47.75%, +31.22%) | **0.3342 (+51.84%, +34.87%)** |
| | | | | | *MRR@20* | | |
| RSC15 | 0.2048 | 0.2305 | 0.2375 | 0.2367 (+15.61%, +2.69%) | 0.3059 (+49.41%, +32.71%) | 0.3045 (+48.70%, +32.08%) | **0.3119 (+52.29%, +35.31%)** |
| VIDEO | 0.2257 | 0.2359 | 0.2609 | 0.2295 (+1.69%, -2.73%) | 0.2970 (+31.63%, +25.92%) | 0.2950 (+30.72%, +25.05%) | **0.3019 (+33.76%, +27.98%)** |
| VIDXL | 0.3740 | 0.3847 | 0.4343 | 0.3608 (-3.53%, -6.21%) | **0.5023 (+34.31%, +30.59%)** | 0.4939 (+32.05%, +28.39%) | 0.5013 (+34.01%, +30.30%) |
| CLASS | 0.0799 | 0.0949 | 0.0995 | 0.0870 (+8.83%, -8.36%) | 0.1176 (+47.14%, +23.90%) | 0.1198 (+49.93%, +26.25%) | **0.1207 (+51.06%, +27.19%)** |

## 4.2 LOSS-FUNCTIONS

We measure the performance gain of the proposed improvements over the baselines. The big accuracy improvement comes from the combination of additional samples and the loss functions (fixed cross-entropy, TOP1-max and BPR-max). Table 2 showcases our most important results. Besides the original version of GRU4Rec and the item-kNN, we included results with cross-entropy (XE) loss without additional sampling to confirm that the fixed cross-entropy loss still performs just slightly better than TOP1. The increase with sampling and the proper loss function is stunning as the best results exceed the accuracy of the original GRU4Rec by $15 - 35\%$ and that of item-kNN by up to $52\%$. BPR-max even performs slightly better $(+1 - 7\%)$ than cross-entropy on 3 of 4 datasets and achieves similar results on the remaining one dataset.

On RSC15, Tan et al. (2016) reported $\sim 0.685$ and $\sim 0.29$ in recall@20 and MRR@20 respectively[11] using data augmentation. Unlike our solutions, data augmentation greatly increases training times. Data augmentation and our improvements are not mutually exclusive, thus it is possible that combining the two methods, even better results can be achieved. A very recent paper Chatzis et al. (2017) proposes the Bayesian version of GRU4Rec and reports $\sim 0.61$ and $\sim 0.25$ in recall@20 and MRR@20 when using 100 units[12]. Therefore our GRU4Rec version is the current best performer so far.

---

[11]Read from figure4. Unfortunately, the results in table1 are for networks trained on various subsets of the training set.

[12]Based on figure1. Their best results (0.6507 and 0.3527) are achieved using 1500 units, which is highly impractical. Even though, our version still performs better wrt. recall when compared to this much bigger network.

Table 3: Results with unified embeddings

| Dataset | Recall@20 | MRR@20 |
|---------|-----------|--------|
| RSC15 | 0.7220 | 0.3070 |
| VIDEO | 0.6612 | 0.2923 |
| VIDXL | 0.8045 | 0.4915 |
| CLASS | 0.3844 | 0.1471 |

### 4.3 UNIFIED ITEM REPRESENTATIONS

Previous experiments did not find any benefits of using an embedding layer before the GRU layers. The role of the embedding layer is to translate item IDs into the latent representation space. In the recommender systems terminology, item embeddings correspond to "item feature vectors". The network has another "item feature matrix" in the form of the output weight matrix. By unifying the representations, i.e. sharing the weight matrix between the embedding layer and the output layer, we learn better item representations quicker. Preliminary experiments (Table 3) show additional improvements in recall@20 and slight decrease in MRR@20 for most of the datasets, however, for the CLASS dataset both recall and MRR are increased significantly when unified embeddings are used ($+15.02\%$ and $+21.87\%$ in recall and MRR respectively, compared to the model trained without embeddings).

## 5 CONCLUSION

We introduced a new class of loss function that together with an improved sampling strategy have provided impressive top-k gains for RNNs for session-based recommendations. We believe that these new losses could be more generally applicable and along with the corresponding sampling strategies also provide top-k gains for different recommendations settings and algorithms such as e.g. matrix factorization or autoencoders. It is also conceivable that these techniques could also provide similar benefits in the area of Natural Language Processing a domain that shares significant similarities to the recommendation domain in terms of machine learning (e.g. ranking, retrieval) and data structure (e.g. sparse large input and output space).

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
