# OpenReview forum: "Recurrent Neural Networks with Top-k Gains for Session-based Recommendations"
_ICLR.cc/2018/Conference — Reject_

### Official Review · AnonReviewer2 · 2017-11-27
**Insightful analysis of existing loss functions, new losses are proposed and provide strong empirical results. Good paper.**

**Rating:** 8
**Confidence:** 4

**Review:**

This is an interesting paper that analyzes existing loss functions for session-based recommendations. Based on the result of these analysis the authors propose two novel losses functions which add a weighting to existing ranking-based loss functions. These novelties are meant to improve issues related to vanishing gradients of current loss functions. The empirical results on two large-scale datasets are pretty impressive.

I found this paper to be well-written and easy to read.  It also provides a nice introduction to some of the recent literature on RNNs for session-based recommendations.

In terms of impact, while it studies a fairly applied (and narrow) question, it seems like it would be of interest to researchers and practitioners in recommender systems.


I have a few comments and questions:

- The results in Figure 3 show that both a good loss function and sampling strategy are required to perform well. This is interesting in the sense that doing the "right thing" according to the model (optimizing using all samples) isn't optimal. This is a very empirical observation and it would be insightful to better understand exactly the objective that is being optimized.

- While BPR-max seems to be the strongest performer (Table 2), cross-entropy (XE) with additional samples is close. This further outlines the importance of the sampling method over the exact form of the loss function.

- In ranking-max losses, it seems like "outliers" could have a bigger impact. I don't know how useful it is to think about (and it is a bit unclear what an "outlier" means in this implicit feedback setting).


Minor comments:

- Around Eq. 4 it may be worth being more explicit about the meaning of i and j.

---

> ### Author Response · Authors · 2017-12-14
> **Thank you for the review**
>
> Thank you very much for the review and the comments. Regarding the decline in performance when performing optimization over all pairs of items, we do believe that this is due to the fact that introducing too many ‘non-relevant’ items in the objective might introduce a negative bias in the learning process.
>
> In the new version of the paper we made the indexes i and j more explicit.

---

### Official Review · AnonReviewer1 · 2017-11-27
**Some interesting insights, but no substantial contributions and probably not well-suited for ICLR**

**Rating:** 4
**Confidence:** 5

**Review:**

This paper presents a few modifications on top of some earlier work (GRU4Rec, Hidasi et al. 2016) for session-based recommendation using RNN. The first one is to include additional negative samples based on popularity raised to some power between 0 and 1. The second one is to mitigate the vanishing gradient problem for pairwise ranking loss, especially with the increased number of negative samples from the first modification. The basic idea is to weight all the negative examples by their “relevance”, since for the irrelevant negatives the gradients are vanishingly small. Experimentally these modifications prove to be effective compared with the original GRU4Rec paper.

The writing could have been more clear, especially in terms of notations and definitions. I found myself sometimes having to infer the missing bits. For example, in Eq (4) and (5), and many that follow, the index i and j are not defined (I can infer it from the later part), as well as N_s (which I take it as the number of negative examples). This is just one example, but I hope the authors could carefully check the paper and make sure all the notations/terminologies are properly defined or referred with a citation when first introduced (e.g., pointwise, pairwise, and listwise loss functions). I consider myself very familiar with the RecSys literature, and yet sometimes I cannot follow the paper very well, not to mention the general ICLR audience.

Regarding the two main modifications, I found the negative sampling rather trivial (and I am surprised in Hidasi et al. (2016) the negatives are only from the same batch, which seems a huge computational compromise) with many existing work on related topic: Steck (Item popularity and recommendation accuracy, 2011) used the same “popularity to the power between 0 and 1” strategy (they weighted the positive by the inverse popularity to the power). More closely, the negative sampling distribution in word2vec is in fact a unigram raised to the power of 0.75, which is the same as the proposed strategy here. As for the gradient vanishing problem for pairwise ranking loss, it has been previously observed in Rendle & Freudenthaler (Improving Pairwise Learning for Item Recommendation from Implicit Feedback, 2014) for BPR and they proposed an adaptive negative sampling strategy (trying to sample more relevant negatives while still keeping the computational cost low), which is closely related to the ranking-max loss function proposed in this paper. Overall, I don’t think this paper adds much on top of the previous work, and I think a more RecSys-oriented venue might benefit more from the insights presented in this paper.

I also have some high-level comments regarding using RNN for session-based recommendation (this was also my initial reaction after reading Hidasi et al. 2016). As mentioned in this paper, when applying RNN on RecSys datasets with longer time-span (which means there can be more temporal dynamics in users’ preference and item popularity), the results are not striking (e.g., Wu et al. 2017) with the proposed methods barely outperforming standard matrix factorization methods. It is puzzling that how RNN can work better for session-based case where a user’s preference can hardly change within such a short period of time. I wonder how a simple matrix factorization approach would work for session-based recommendation (which is an important baseline that is missing): regarding the claim that MF is not suited for session-based because of the absence of the concept of a user, each session can simply be considered as a pseudo-user and approaches like asymmetric matrix factorization (Paterek 2007, Improving regularized singular value decomposition for collaborative filtering) can even eliminate the need for learning user factors. ItemKNN is a pretty weak baseline and I wonder if a scalable version of the SLIM (Ning & Karypis 2011, SLIM: Sparse Linear Methods for Top-N Recommender Systems) would give better results. Finally, my general experience with BPR-type of pairwise ranking loss is that it is good at optimizing AUC, but not very well-suited for head-heavy metrics (MRR, Recall, etc.) I wonder how the propose loss would perform comparing with more competitive baselines.

Regarding the page limit, given currently the paper is quite long (12 pages excluding references), I suggest the authors cutting down some space. For example, the part about fixing the cross entropy is not very relevant and can totally be put in the appendix.

Minor comment:

1. Section 3.3.1, “Part of the reasons lies in the rare occurrence…”, should r_j >> r_i be the other way around?

---

> ### Author Response · Authors · 2017-12-14
> **Thank you for the review**
>
> Thank you for the extensive review. We tried to fix some of the notation issues you mentioned in the paper and uploaded a new version. Regarding the ranking losses we added a note when pointwise or listwise loss ranking losses are first mentioned to clarify them, we believe that much of the ML community is familiar with these terms as they originate from the learning to rank literature and are not specific to the RecSys community.
>
> While several negative sampling strategies have been proposed we do believe that ours is quite different from past ones given that we also adapt the loss functions (BPR and TOP1) to the proposed sampling strategies by using a relaxation of the max operator (softmax) to adjust the loss (and gradients).
> None of the previously proposed sampling schemes use this. Rendle (2014) focuses on finding highly scored negative items for the BPR loss while Steck (2011) weight the sampled items with their popularity. Softmax over the scores in the loss function has, to the best of our knowledge, not been used before. Note also that the massive performance benefits we find in the experiments result from the combined use of the new losses and the proposed sampling strategy.
>
> Regarding the high level comments about RNNs in session-based recommendation, both in the original Hidasi et.al (2016) but also in other works extending upon Hidasi et. al. (2016) e.g. Chatzis et. al. (2017) matrix factorisation approaches (BPR-MF to be precise)  have been used as baselines in the way you describe i.e. the individual sessions are treated as users in the matrix. The question that arises is how to then serve recommendations, and in the original Hidasi et.al (2016) the item factors are used to compute similarities with the last clicked item and in the Chatzis et. al. (2017) the item factors of the session (so far) are averaged. Both approaches lead to very poor results even compared to item-knn (and an order of magnitude below RNN), both are reported in the papers.
> Neither asymetric MF approaches (which essentially use averaging or SLIM which uses weighting of the factors(and would need to be rerun at every click) can improve upon these results enough so that MF becomes competitive with RNNs.
>
> In fact the Wu et. al. (2016) paper (Recurrent Recommender Nets)  does something similar in that it uses RNNs to perform a type of matrix factorization whereby two RNNs are used (one on the user side and one on the item side) to predict user/item factors given the items that the user has seen so far and the users that have seen the item so far.
> They also evaluate with RMSE (which correlates poorly with IR metrics) on the Netflix data, note that the Netflix data is not a session-based dataset and matrix factorization approaches have been shown to perform well there. As a last twist note also that Wu et. al. (2016) initialize these RNNs by performing a PMF (to learn the initial factors of the matrix factorization)  so it is unclear if and to what extend the RNNs actually learn anything from the data. It is thus no surprise to us that the results in Wu et. al. (2016) are quite poor as the RNNs are used in a rather convoluted way (to predict factors of a matrix factorization instead of the next item) and on data that is not strongly sequential or session-based.
>
> Regarding BPR performance on MRR and Recall our experience is that it performs actually quite well in these metrics.

---

### Official Review · AnonReviewer3 · 2017-11-29
**This paper discussed the issues for optimizing the loss functions in GRU4Rec and proposed tricks for optimize the loss functions, and also proposed enhanced version of the loss functions for GRU4Rec. Good performance improvements have been reported for the several datasets to show the effectiveness of the proposed methods.**

**Rating:** 6
**Confidence:** 5

**Review:**

This paper discussed the issues for optimizing the loss functions in GRU4Rec and proposed tricks for optimize the loss functions, and also proposed enhanced version of the loss functions for GRU4Rec. Good performance improvements have been reported for the several datasets to show the effectiveness of the proposed methods.

The good point of this work is to show that the loss function is important to train a better classifier for the session-based recommendation. This work is of value to the session-based recommendations.

Some minor points:
I think it may be better if the authors could put the results of RSC15 from Tan (2016) and Chatzis ect. (2017) into table 2 as well.
As these work has already been published and should be compared with and reported in the formal table.

---

> ### Author Response · Authors · 2017-12-14
> **Thank you for the review**
>
> Thank you for the review,
>
> Regarding the minor comment,
> we did not put the Tan (2016) and Chatzis (2017)  results directly in the table because the experimental protocols are slightly different, we generally avoid presenting results in the same table that are not run under the same experimental setting (test-train splits etc.) Notice though that the relative increase over the baseline GRU that we report is way higher than the results in Chatzis (2017) and equal to Tan (2016) (but with way less computational cost).

---

### Decision · Program_Chairs · 2018-01-29
**ICLR 2018 Conference Acceptance Decision**

**Decision:**

Reject

**Comment:**

While the use of RNNs for building session-based recommender systems is certainly an important class of applications, the main strength of the paper is to propose and benchmark practical modifications to prior RNN-based systems that lead to performance improvements. The reviewers have pointed out that the writing in the paper needs improvement,  modifications are somewhat straightforward and some expected baselines such as comparisons against state of the art matrix-factorization based methods is missing. As such the paper could benefit from a revision and resubmission elsewhere.